# How Relevant Is the Parallax Effect on Low Centered Pelvic Radiographs in Total Hip Arthroplasty

**DOI:** 10.3390/jpm13060881

**Published:** 2023-05-23

**Authors:** Markus Weber, Matthias Meyer, Frederik Von Kunow, Bernd Füchtmeier, Axel Hillmann, Christian Wulbrand

**Affiliations:** 1Faculty of Medicine, University of Regensburg, 93053 Regensburg, Germany; 2Department of Orthopedic and Trauma Surgery, Barmherzige Brueder Regensburg Medical Center, 93047 Regensburg, Germany; 3Department of Orthopedic Surgery, Regensburg University Medical Center, 93077 Bad Abbach, Germany

**Keywords:** total hip arthroplasty, cup position, parallax effect

## Abstract

The correct cup position in total hip arthroplasty (THA) is usually assessed on anteroposterior low centered pelvic radiographs, harboring the risk of misinterpretation due to projection of a three-dimensional geometry on a two-dimensional plane. In the current study, we evaluate the effect of this parallax effect on the cup inclination and anteversion in THA. In the course of a prospective clinical trial, 116 standardized low centered pelvic radiographs, as routinely obtained after THA, were evaluated regarding the impact of central beam deviation on the cup inclination and anteversion angles. Measurements of the horizontal and vertical beam offset with two different methods of parallax correction were compared with each other. Furthermore, the effect of parallax correction on the accuracy ofmeasuring the cup position was investigated. The mean difference between the two parallax correction methods was 0.2° ± 0.1° (from 0° to 0.4°) for the cup inclination and 0.1° ± 0.1° (from −0.1° to 0.2°) for the anteversion. For a typically intended cup position of a 45° inclination and 15° anteversion, the parallax effect led to a mean error of −1.5° ± 0.3° for the inclination and 0.6° ± 1.0° for the anteversion. Central beam deviation resulted in a projected higher cup inclination up to 3.7°, and this effect was more prominent in cups with higher anteversion. In contrast, the projected inclination decreased due to the parallax effect up to 3.2°, especially in cups with high inclination. The parallax effect on routinely obtained low centered pelvic radiographs is low and not clinically relevant due to the compensating effect of simultaneous medial and caudal central beam deviation.

## 1. Introduction

Correct positioning of the acetabular cup is crucial for short- and long-term outcomes after total hip arthroplasty (THA) [1]. Failure can result in impingement, instability and polyethylene wear [2,3,4]. In clinical practice, the inclination and anteversion of the acetabular component are usually assessed on anteroposterior low centered pelvic radiographs. However, measurements on plain radiographs might be susceptible to error since a three-dimensional geometry is projected on plain film. In the literature, differences regarding cup inclination and anteversion of over 20° have been reported when compared with 3D-CT [5]. One source of this potential deviation is the so-called parallax effect due to deviation of the X-ray beams [6]. Although several methods have been described in the literature to correct for the central beam offset [6,7], the clinical relevance of this effect on postoperative low centered anteroposterior pelvic radiographs is unknown.

In the present study, we aim to analyze the effect of the central beam offset on the cup inclination and anteversion angles as measured on postoperative low centered pelvic radiographs.

## 2. Subjects

In the course of a registered, prospective controlled trial (DRKS00000739, German Clinical Trials Register), low centered anteroposterior pelvic radiographs were obtained postoperatively from patients undergoing THA. This investigation was approved by the local ethics commission (No. 10-121-0263). The current study is a secondary analysis of a larger project [8]. The primary outcome of this study was to assess whether the artificial joint’s range of motion could be improved with a computer-assisted, functional optimization of the cup position and cup containment. A consecutive series of 783 patients with osteoarthritis of the hip was screened. The inclusion criteria were the following: age between 50 and 75 years, an American Society of Anesthesiologists (ASA) score ≤ 3, unilateral osteoarthritis of the hip (up to Kellgren 2 on the contralateral side), no prior hip surgery and no hip dysplasia or trauma. In total, 597 patients did not meet the inclusion criteria, 27 patients declined participation in the study, and another 19 patients could not be included for other reasons (e.g., cancellation of the operation or increased inflammatory factors during a blood examination the day before surgery). In total, a consecutive series of 135 patients was enrolled in this single center study. Nineteen patients had to be withdrawn from the current study. Four patients withdrew their informed consent and refused further participation in the study as well as use of their data. In seven data sets, no postoperative CT was available, and in eight cases, there was an obvious rotational error on the postoperative radiograph. In total, the records of 116 THAs were included for final analysis (Figure 1). The characteristics of the study group are shown in Table 1.

After giving written consent, THA was performed by four senior arthroplasty surgeons at the University Arthroplasty Center. All operations were performed in the lateral decubitus position through a minimally invasive antero-lateral surgical approach to the hip joint, following an intermuscular and interneural tissue plane between the tensor muscle and the gluteus medius muscle [9]. In all cases, the same press-fit components (Pinnacle; DePuy, Warsaw, IN, USA) and the same cement-free hydroxyapatite-coated stems (Corail; DePuy, Warsaw, IN, USA) were used. The tribological pairing consisted of polyethylene liners and metal heads with a diameter of 32 mm.

After surgery, low centered anteroposterior pelvic radiographs were obtained in a standing position at full weight-bearing (MULITX TOP ACSS, Siemens, Erlangen, Germany). The radiographic technician made sure that the pelvis was set parallel to the plane of the film and the leg was placed in a neutral position without rotation or flexion of the hip joint. The central beam was directed at the symphysis. All radiographs were taken under these highly standardized conditions (focus-film distance of 115 cm, 75 kV, automatic exposure). However, in eight cases, a correct placement was not achievable, and these radiographs were excluded. At the same time, a CT scan was made from the pelvis down to the femur condyles (Somatom Sensation 16; Siemens, Erlangen, Germany). Operative characteristics of the study group are shown in Table 2.

Radiographic measurements were obtained with the help of digital planning software (MediCAD, Hectec, Germany) for all patients. The radiographic magnification was corrected by using the implanted head of the implant as a scaling object with a known diameter of 32 mm. The central beam was located using the intersection of two crossing lines from the opposing picture edges. The vertical and horizontal central beam offset were defined as the vertical and horizontal distance of the central beam to the center of the cup. Radiographic measurement of the cup inclination and anteversion was carried out according to Lewinnek [10] (Figure 2). We then calculated the parallax effect using the correction of Derbyshire [6] and Schwarz [7]. The link between the two methods is TP = cos(arctan(b/a) + [(90 − inclination) × π/180]) × (a^2^ + b^2^)^1/2^, where a represents the horizontal beam offset and b is the vertical beam offset (Appendix A). CT scans were used to rule out a retroversion of the implant. The effect size of the parallax correction was analyzed, and the two methods of Derbyshire and Schwarz were compared with each other. Furthermore, the cup positions were changed according to the recommendations of Lewinnek’s safe zone, varying between 30° to 50° in inclination and 5° to 25° in anteversion [11] in the radiographic definition [12] for each image. Then, we compared the parallax effect due to the individual central beam deviation of each image in relation to the estimated different cup positions. Positive values represent an underestimation due to the parallax effect (lower anteversion or inclination), and correspondingly, negative values represent an overestimation (higher inclination and anteversion). For statistical analysis, continuous data are presented as the mean ± standard deviation (range from max to min). Regression analyses were performed to check for correlations of the magnitude of the parallax effect and anthropometric characteristics such as BMI, sex, age and treatment side. IBM SPSS Statistics 25 (SPSS Inc., Chicago, IL, USA) was used for analysis.

## 3. Results

When analyzing the deviation of the central beam in relation to the cup’s center of rotation, a mean horizontal offset of 7.8 ± 1.2 cm was measured on the low centered pelvic radiographs after THA. The corresponding mean vertical offset amounted to 5.9 ± 2.3 cm (Figure 3). This resulted in a mean horizontal central beam offset angle of 4.0° ± 0.6° and mean vertical offset angle of 3.0° ± 1.2°. The parallax correction according to Derbyshire resulted in a change of −1.7° ± 0.6° (from −3.9° to −0.3°) for the cup inclination and 0.6° ± 1.1° (from −2.2° to 3.4°) for the cup anteversion. Similarly, parallax correction to Schwarz led to a mean change of −1.9° ± 0.7° (from −4.3° to −0.3°) for the inclination and of 0.6° ± 1.1° (from −2.4° to 3.5°) for the anteversion. The mean difference of the two correction methods was 0.2° ± 0.1° (from 0° to 0.4°) for the inclination and 0.1° ± 0.1° (from −0.1° to 0.2°). Individual differences between the two methods are shown in Figure 4.

For a typically intended cup position of 45° inclination and 15° anteversion according to the radiographic definition of Murray, the parallax effect would lead to a mean deviation in the measuring cup position of −1.5° ± 0.3° (from −2.2° to −0.9°) for the inclination and 0.6° ± 1.0° (from −1.8° to 2.9°) for the anteversion. Ninety-five percent of the individual deviations were located in an interval from −2.1° to −0.9° for the inclination and from −1.4° to 2.6° for the anteversion. When varying the cup position according to the recommendations of Lewinnek’s safe zone between 30° and 50° inclination and 5° and 25° anteversion, the parallax effect resulted in a projected higher cup inclination by up to 3.7°and cups with higher anteversion. In contrast, the projected anteversion decreased due to the parallax effect by up to 3.2°, especially in cups with high inclination (Table 3).

A correlation between the extent of the parallax effect and the patient’s sex was observed (*p* ≤ 0.002). The BMI showed only a weak correlation regarding inclination (*p* = 0.01). Other anthropometric characteristics such as age and treatment side showed no association with the parallax effect (Table 4).

## 4. Discussion

Correct cup position is crucial for the range of motion and outcome after THA. The target areas for a cup’s position have been identified in literature [11], but the accuracy of measuring the cup position is still a matter of debate since, in clinical practice, plain radiographs of the pelvis are commonly used for assessment. However, this is a potential source of error due to the divergence of the X-ray beams, resulting in the so-called parallax effect. The clinical relevance of the parallax effect has been previously shown in spine surgery [13] and corrective osteotomy [14]. Therefore, we aimed to evaluate the impact of this parallax effect on the accuracy of cup position measurements in THA on low centered pelvic radiographs. Our results revealed a negligible effect on low centered radiographs, with a mean error of 1–2° for a cup’s inclination and anteversion. Both correction methods described in the literature according to Derbyshire and Schwarz were comparable, with a mean difference of less than 0.4°.

The present study revealed that the deviation of the central beam in relation to the cup’s center of rotation averaged 7.8 cm regarding the horizontal offset and 5.9 cm regarding the vertical offset, as measured on low centered pelvic radiographs. The images were obtained in the radiographic unit for routine postoperative follow-ups. This led to a mean horizontal central beam offset angle of 4.0° and a mean vertical offset angle of 3.0°. These values differ from the settings used by Derbyhire in the model, which were set to a medial deviation of 11 cm and inferior deviation of 4.5 cm [6].

In the literature, the accuracy of the parallax correction according to Derbyshire was reported within a few hundredths of a degree for the cup anteversion and within 0.2° for the cup inclination [6]. We compared this method with another method described by Schwarz et al. [7] and found a high correlation, with a difference below 0.2° for both the cup inclination and anteversion. Without correction, the parallax effect resulted in an overestimation of the cup inclination, with a mean of about 2°, and an underestimation of the cup anteversion, with a mean of about 1°. Therefore, the effect size of the parallax effect is within the same range of 2° as the measurement error of defining the inclination or anteversion angle on radiographs [7]. In contrast, when using intraoperative fluoroscopy, the focus-film distance and, depending on the applied positioning of the patient on the table, the patient-film distance can differ significantly from the standard values we know from routine image acquisition postoperatively in the radiographic unit. Therefore, the parallax effect is of greater relevance for radiographs obtained during the operation and should be taken into account by using a standardized intraoperative technique to minimize these effects [15].

We then assessed the parallax effect for different cup positions with varying cup inclinations between 30° and 50° and cup anteversions between 5° and 25° in relation to the safe zone recommended by Lewinnek [11]. Although recent studies have shown that the historical target values for cup position did not guarantee a free range of motion without dislocation [16], we chose theses values because they are still frequently used in daily practice. In this simulation, the cup anteversion was underestimated by up to 4°, and the cup inclination was similarly overestimated by up to 4°. The error for the cup inclination was higher in cups with higher anteversion. These results are in line with Derbyshire, who similarly reported a deviation of up to 4° regarding the cup inclination and anteversion in his model. Correspondingly, the parallax effect resulted in an underestimation of the cup anteversion and overestimation of the cup inclination [6]. This should be taken into account when measuring the cup position in the daily routine.

When researching the anthropometric characteristics associated with the effect size of the parallax effect, we found the strongest correlation with the patient’s sex. This sex-specific difference might be related to differences in the anatomic configuration of the pelvis between men and women. Sex-related differences of the pelvic anatomy have been shown in a previous study dealing with psoas impingement [17]. However, the detailed relation between sex and the parallax effect remains unclear.

There are several limitations regarding this study. First, the results rely on the imaging acquisition method of our hospital as described in the Methods section and might differ in other settings. In particular, when using intraoperative fluoroscopy, the focus-film and object-film distances differ from the settings applied in this study, and thus the parallax effect can be of greater relevance. For further research, a verification study with a skeletal pelvic model and acetabular cup is recommended. Second, we only focused on one source of error for measurement of the cup position. Other parameters such as the methodology of assessing anteversion [18], pelvic tilt and pelvic rotation [19] further affect the accuracy of cup position evaluation. Third, only one type of acetabular component was used in the study (Pinnacle; DePuy, Warsaw, IN, USA). However, it seems unlikely that a different cup design will lead to different findings. Fourth, the applied approach (anterior vs. postero-lateral) as well as the position of the patient (lateral decubitus versus supine) influence the range of cup inclination and anteversion. Fifth, the optimal cup position is still under debate. In addition to single target areas for the cup position [11], modern concepts concentrate on combined version angles of the cup and stem [20]. Furthermore, functional aspects such as pelvic tilt have become more relevant [19]. All in all, the question of the “perfect” cup position has not been completely answered yet. This raises the question of why we deal with the accuracy of measurement methods within single degrees if the target area is still unknown.

In conclusion, due to the simultaneously medial and inferior deviation of the X-ray beams on low centered pelvic radiographs, the parallax effect is usually negligible, with mean deviations below 2°. For correction, different methods with similar accuracy are available. The parallax effect leads to an underestimation of the cup anteversion and overestimation of the cup inclination.

## Figures and Tables

**Figure 1 jpm-13-00881-f001:**
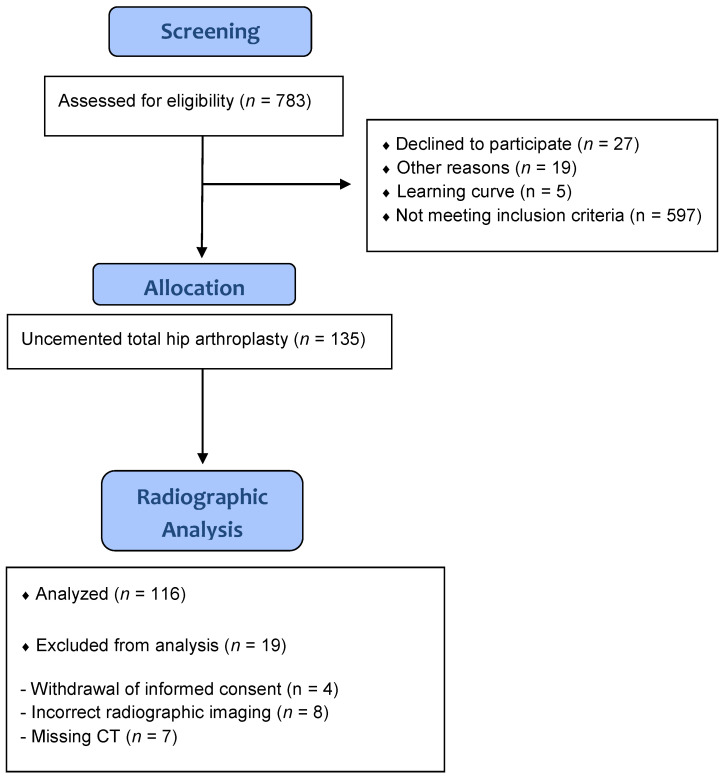
Flow diagram of the study group.

**Figure 2 jpm-13-00881-f002:**
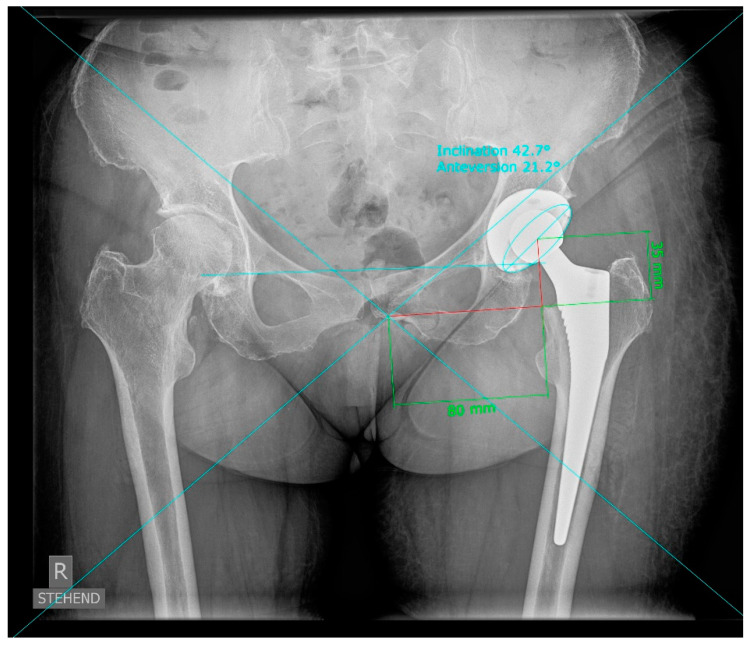
Radiographic measurement of horizontal and vertical central beam offset on anteroposterior low centered pelvic radiographs.

**Figure 3 jpm-13-00881-f003:**
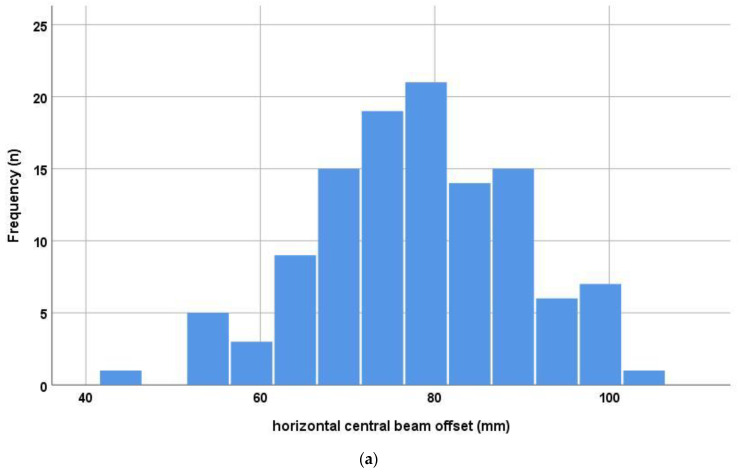
Histograms of horizontal (**a**) and vertical (**b**) central beam offset in the study group.

**Figure 4 jpm-13-00881-f004:**
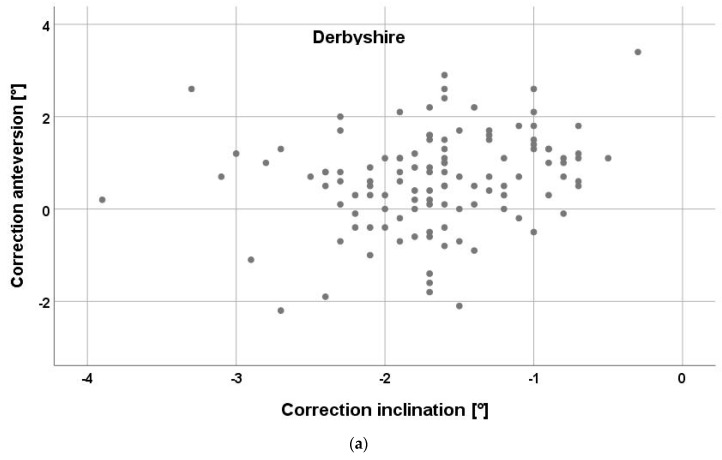
Scatter plot of the impact of parallax effect on cup inclination and anteversion according to Derbyshire (**a**) and Schwarz (**b**) correction.

**Table 1 jpm-13-00881-t001:** Anthropometric characteristics of the study group.

Study group	number (*n*) = 116
Gender (female/male)	64/52
Age (years)	62.7 ± 7.6
BMI (kg/m^2^)	27.0 ± 4.1
Treatment side (left/right)	53/63
ASA 1	24 (20.7%)
ASA 2	60 (51.7%)
ASA 3	32 (27.6%)

ASA = American Society of Anesthesiology score. Categorial data values are given as relative and absolute frequencies, and quantitative data values are given as mean ± standard deviation.

**Table 2 jpm-13-00881-t002:** Operative characteristics of the study group.

Cup size	54 (48–62)
Femoral component size	12 (9–16)
Operation time (minutes)	67.3 ± 13.8
CT Cup inclination APP radiographic (°)	42.6 ± 5.9
CT Cup anteversion APP radiographic (°)	18.1 ± 8.1
X-ray Cup inclination APP radiographic (°)	44.6 ± 5.3
X-ray Cup anteversion APP radiographic (°)	18.7 ± 6.0

Categorial data values are given as relative and absolute frequencies, and quantitative data values are given as mean ± standard deviation or median (range).

**Table 3 jpm-13-00881-t003:** Estimated difference between cup inclination and anteversion due to parallax effect for different cup positions according to Lewinnek’s safe zone.

**Cup**	**30_5**		**40_5**		**50_5**	
	**Anteversion**	**Inclination**	**Anteversion**	**Inclination**	**Anteversion**	**Inclination**
MW	−0.6	−0.5	0.2	−0.6	1.1	−0.6
STD	1.1	0.1	1.1	0.1	1.0	0.1
Min	−3.6	−0.7	−2.4	−0.8	−1.2	−0.9
Max	1.9	−0.3	2.6	−0.3	3.2	−0.3
**Cup**	**30_15**		**40_15**		**50_15**	
	**Anteversion**	**Inclination**	**Anteversion**	**Inclination**	**Anteversion**	**Inclination**
MW	−0.7	−1.4	0.2	−1.5	1.1	−1.5
STD	1.1	0.2	1.1	0.2	1.0	0.3
Min	−3.7	−1.9	−2.5	−2.1	−1.3	−2.2
Max	1.9	−1	2.6	−0.9	3.2	−0.8
**Cup**	**30_25**		**40_25**		**50_25**	
	**Anteversion**	**Inclination**	**Anteversion**	**Inclination**	**Anteversion**	**Inclination**
MW	−0.7	−2.4	0.2	−2.5	1.0	−2.5
STD	1.1	0.3	1.1	0.4	1.0	0.4
Min	−3.8	−3.2	−2.6	−3.5	−1.3	−3.7
Max	1.8	−1.7	2.5	−1.5	3.1	−1.3

**Table 4 jpm-13-00881-t004:** Multivariable regression of anthropometric characteristics and effect size of parallax effect on cup inclination and anteversion.

**Inclination**	**Coefficient**	**95% Confidence Interval**		***p* Value**
Sex	0.30	0.06	0.25	0.002
Age	0.05	0.00	0.01	0.6
BMI	0.07	−0.01	0.02	0.5
Treamtment Side	−0.10	−0.14	0.04	0.3
**Anteversion**	**Coefficient**	**95% Confidence Interval**		***p* Value**
Sex	0.33	0.31	1.0	<0.001
Age	−0.01	−0.02	0.2	1.0
BMI	0.23	0.01	0.14	0.01
Treamtment Side	0.07	−0.22	0.49	0.4

## Data Availability

The data presented in this study are available on request from the corresponding author. The data are not publicly available due to privacy restrictions.

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
