# Peer review of "How Relevant Is the Parallax Effect on Low Centered Pelvic Radiographs in Total Hip Arthroplasty"

_jpm, 2023, doi:10.3390/jpm13060881_

Round 1

Reviewer 1 Report

It seems that there are no real values of the cup inclination and anteversion in individual clinical cases. Therefore, it is quite hard to appreciate the scientific significance. However, this article is very suggestive to the clinical evaluation of Total Hip Arthroplasty. And several parts should be improved according to the review comments. For further research, verification study with the skeletal pelvic model and acetabular cup is recommended as the part of discussion.

Line 43. Heading of Materials or Subjects is required. THA cases should be described as the section of Materials or Subjects rather than Methods.

 Line 96. In the equation of TP, the parameter of d is not defined. It might be the focus-film distance.

 Line118, 128 (All figures) . The figure title and captions should be places under the figure.

Figure 1 & 3 are not indicated in the main text.

Figure 2. The blue and green letters in the x-ray image are unclear, which should be changed to white color with black shadow. 

The values with the round brackets described in the main should be kindly explained as the value range, i.g. mean ± s.d. (max to min).

Author Response

We thank the reviewer for this profound review and have included a detailed point by point analysis as attachment.

Reviewer 2 Report

Dear Authors,

I read your paper with interest and appreciation. Below are some remarks I hope will improve the manuscript. 

In my opinion you should mention in introduction and/or discussion about the influence of the approach (anterior vs. postero-lateral) as well as the position of the patient (lateral decubitus vs. on the back) on the range of cup inclination and anteversion. The same remarks should be mentioned in the limitations of your study. 

Author Response

We thank the reviewer for this profound review and have included a detailed point by point analysis.
